# Association of Lumbar Paraspinal Muscle Morphometry with Degenerative Spondylolisthesis

**DOI:** 10.3390/ijerph18084037

**Published:** 2021-04-12

**Authors:** Eun Taek Lee, Seung Ah Lee, Yunsoo Soh, Myung Chul Yoo, Jun Ho Lee, Jinmann Chon

**Affiliations:** 1Department of Physical Medicine & Rehabilitation, College of Medicine, Kyung Hee University, Seoul 02447, Korea; mylot91@naver.com (E.T.L.); Isarang80@gmail.com (S.A.L.); soyuns@gmail.com (Y.S.); famousir@naver.com (M.C.Y.); 2Department of Neurosurgery, College of Medicine, Kyung Hee University, Seoul 02447, Korea; moo9924@khu.ac.kr

**Keywords:** degenerative spondylolisthesis, polyradiculopathy, erector spinae, multifidus, paraspinal muscles, fatty degeneration, cross sectional area, lumbar

## Abstract

The objective of this study was to assess the cross-sectional areas (CSA) of lumbar paraspinal muscles and their fatty degeneration in adults with degenerative lumbar spondylolisthesis (DLS) diagnosed with chronic radiculopathy, compare them with those of the same age- and sex-related groups with radiculopathy, and evaluate their correlations and the changes observed on magnetic resonance imaging (MRI). This retrospective study included 62 female patients aged 65–85 years, who were diagnosed with lumbar polyradiculopathy. The patients were divided into two groups: 30 patients with spondylolisthesis and 32 patients without spondylolisthesis. We calculated the CSA and fatty degeneration of the erector spinae (ES) and multifidus (MF) on axial T2-weighted magnetic resonance (MR) images from the inferior end plate of the L4 vertebral body levels. The functional CSA (FCSA): CSA ratio, skeletal muscle index (SMI), and MF CSA: ES CSA ratio were calculated and compared between the two groups using an independent t-test. We performed logistic regression analysis using spondylolisthesis as the dependent variable and SMI, FCSA, rFCSA, fat infiltration rate as independent variables. The result showed more fat infiltration of MF in patients with DLS (56.33 vs. 44.66%; *p* = 0.001). The mean FCSA (783.33 vs. 666.22 mm^2^; *p* = 0.028) of ES muscle was a statistically larger in the patients with DLS. The ES FCSA / total CSA was an independent predictor of lumbar spondylolisthesis (odd ratio =1.092, *p* = 0.016), while the MF FCSA / total CSA was an independent protective factor (odd ratio =0.898, *p* = 0.002)

## 1. Introduction

Degenerative lumbar spondylolisthesis (DLS) is a disorder that causes the slip of one vertebral body (VB) over the other one below due to degenerative changes in the spine. The reported incidence of DS varies among studies, from 4.1% in cadaveric material, 13.6% in a population-based cohort study, and up to 28.6% in a clinical cohort study [1,2]. DLS is the major cause of spinal canal stenosis and is often associated with back and leg pain [3].

There are some known characteristics of DLS. The male-to-female ratio of DLS was 1:5–6. DLS is uncommon in patients aged < 50 years. The most commonly affected location is the L4/L5 level, and the slip rarely exceeds 30% of the width of the subjacent vertebra [4]. The possible causative factors of DLS are the degree of pelvic inclination and declination, iliolumbar ligament strength, degree of adjacent disk or facet joint degeneration, and relative L1–S1 angle [5]. Patients with DLS are usually asymptomatic, but the condition may progress to spinal stenosis [4]. 

Degenerative spondylolisthesis is associated with increased age in both sexes [4]. Aging leads to skeletal muscle mass atrophy, which can be replaced by connective tissues such as fat. These changes, known as sarcopenia, are due to a reduction in both the number and size of muscle fibers. 

The lumbar muscles functionally support and maintain spine stability [6]. The multifidus muscle (MF) is the most medial part of the lumbar paraspinal muscles, innervated by the dorsal ramus medial branch of the segmental nerve [7]. The erector spinae muscle (ES) acts as a global muscle affecting movement and muscular strength [6]. During the last decade, an increasing number of studies have further explored the interaction among the paraspinal muscles, low back pain (LBP), and spinal pathology. Degeneration of the paraspinal muscle with decreased isometric force of the L-spine, low back pain, disk herniation, polyradiculopathy, and DLS have been reported in many studies measuring the cross-sectional area (CSA) of the paraspinal muscles [6,8,9,10,11]. Spinal stability is an important factor to consider in the evaluation and treatment of patients with DLS [12]. Therefore, evaluation of CSA of paraspinal muscle can be crucial because these muscles can directly influence segmental stability and control of the lumbar spine owing to their attachment to the spinal column.

In this study, the patient population comprised only women, aged 65–85 years, as age-related changes in skeletal muscle quality differ between the sexes [4]. Men have less fatty degeneration of paraspinal muscles compared to women; therefore investigation of fat infiltration in women may be needed first for a more accurate comparison. We hypothesized that the deficit in muscular stabilization caused by degeneration of the MF and ES is associated with DLS. However, when patients with DLS have chronic radiculopathy or disk degeneration, it is difficult to differentiate the correlation between MF atrophy. Electromyography assess denervation in the multifidus muscle more accurately and perhaps contribute to the acquisition of additional evidence to support the correlation between multifidus muscle fatty infiltration and structural parameters. Therefore, the current study assessed the CSA of the lumbar paraspinal muscles and degree of fat infiltration into the MF and ES in adults with DLS and chronic radiculopathy. We aimed to determine the association between DLS and CSA of the paraspinal muscle by precluding the factors that can influence the CSA of the paraspinal muscle.

## 2. Methods

### 2.1. Subjects

We, retrospectively, analyzed 83 women aged 65–85 years, who were diagnosed with lumbar polyradiculopathy on electrodiagnostic study from January 2017 to April 2020 at the clinic of the Department of Rehabilitation Medicine of Kyung Hee University Hospital. The inclusion criteria were: (1) female sex, (2) an age of 65–85 years, (3) presence of L-spine MRI study, and (4) polyradiculopathy at the L-spine level. The electrodiagnostic criteria for radiculopathy were abnormal spontaneous activity, abnormal motor unit morphology consistent with nerve injury (polyphasic, large amplitude, increased duration), and reduced recruitment patterns in muscles innervated by the same myotome but different muscles. Patients with polyneuropathy, previous lumbosacral spinal surgery, systemic disorders, spinal fractures, tumors, or infections were excluded. Approval for this study was obtained from the institutional review board (IRB; IRB number: 2020-12-048).

In all, 83 patients were eligible based on their electrodiagnostic findings. All patients had radiculopathy between the L3 and S1 levels, and 17 patients had unilateral polyradiculopathy (22.6%). A total of nine patients had an ongoing denervation pattern; another 21 patients were excluded from the study. Finally, 62 patients were selected and divided into two groups based on their radiologic findings: patients with spondylolisthesis (30 cases) and those without spondylolisthesis (32 cases) (Figure 1).

### 2.2. Measurements and Procedures

All patients underwent lumbosacral spine magnetic resonance imaging (MRI). MRI and electromyography study days were not separated by more than 6 weeks. The degree of spondylolisthesis was determined according to the Meyerding classification. Disk degeneration was graded on a 5-point ordinal scale, as described by Pfirrmann et al. [13]. Facet arthropathy at the level of the listhesis was recorded as a categorical variable based on the established MRI criteria [14]. Axial T2-weighted MRI images were obtained at the level of the L4 inferior endplate. At the L4/L5 disk level, the anterior and posterior middle disk heights were measured to evaluate the height disk [15]. The CSAs of both sides of the MF and ES and L4 inferior endplate margin were measured by drawing their outlines with the region of interest (ROI) using a PiView program (Infinitt, Seoul, Korea). The facet arthropathy grade was measured. 

Axial T2-weighted MR images were exported from a picture archiving and communication system (PACS) workstation and reformatted using image processing software (Image J, version 1.53e, National Institutes of Health, Wayne Rasband, Washington, DC, USA). Quantitative measurements of the erector spinae and multifidus muscles were obtained for each patient using a method previously proposed by Fortin et al. [16]. 

CSA was defined by manually tracing the fascial boundary of the multifidus and erector spinae bilaterally. Functional CSA (FCSA) measurements were obtained using a highly reliable thresholding technique, which is based on the difference in signal intensity between muscle (low signal) and fat tissue (high signal), allowing for the separation of both tissues. Total CSA and FCSA were measured individually for the MF and ES muscles. The reliability of the FCSA and total CSA measurements was relatively equivalent across all spinal levels [17]. FCSA was measured by selecting a threshold signal intensity within the total muscle CSA to include only pixels within the lean muscle range (Figure 2). The grayscale range for lean muscle tissue was selected for every patient in each slice. The six sample ROIs within the bilateral paraspinal muscle were taken from areas of lean muscle tissue visible on each slice, avoiding the inclusion of any visible pixel of fat (Figure 2). The maximum signal intensity acquired from the six ROIs was selected as the highest threshold to distinguish muscle tissue from fat. Subsequently, the percentage of red area (ratio of FCSA to total CSA) in the muscle compartment was calculated (Figure 2). The fat infiltration rate can be calculated (100%- ratio of FCSA to total CSA%).

The previous two studies showed multifidus atrophy and erector spinae hypertrophy in patients with DLS, we first calculated the MF: ES ratio as a predictive factor for DLS [18,19]. Previous studies have used the relative CSA (rCSA) to compensate for the influence of individual body shape, body weight, and height on the CSA of the muscles [6]. Therefore, we calculated the rCSA of the muscles, which is defined as the ratio of the CSA of muscles to that of the lower margin of the L4 vertebra. In another previous study on paraspinal lean muscle measurement, the skeletal muscle index (SMI) was used to normalize values across varying patient heights, akin to the body mass index (BMI) calculation [20]. For example, SMI of total CSA = total CSA of muscle on an axial scan (expressed as cm^2^) divided by the square of the patient’s height (expressed as cm^2^).

All CSA measurements were performed independently twice, with a 1-week interval, by the first author (E.T.L.) to minimize the potential for error in constructing the polygons around the margins of muscles. The average values of the two measurements were used for statistical analysis. The clinical and electrodiagnostic findings and DLS group of all cases were blinded. The measurement of CSA, disk degeneration grade, facet arthropathy grade, disk height were performed by the first author (E.T.L.) with 4 years of experience in spinal MRI measurement.

### 2.3. Statistical Analysis

Statistical analysis was performed using the SPSS ver. 25.0 for Windows (IBM Corp., Armonk, NY, USA). Cohen’s effect size and power was calculated using G Power ver.3.1. We performed an independent t-test and Fisher’s exact test to analyze the differences in the demographic and radiologic differences. An independent t-test was used to investigate statistical relationships of total CSA, rCSA, FCSA, FCSA/total CSA (%), MF: ES ratio, and SMI between the two groups. These variables were entered into binary logistic regression analysis. To determine which variables were the most statistically appropriate independent predictors (largest area under the curve (AUC) area) of DLS, we performed five logistic regression analyses. We used DLS as the dependent variable and five sets of variables (MF FCSA/ES FCSA, SMI FCSA, ratio of FCSA to total CSA, FCSA, rFCSA) as the independent variables. BMI, age, disk height, facet arthropathy grade, disk degeneration grade were adjusted in each logistic regression analysis. Statistical significance was set at *p*-values less than 0.05. All values were presented as mean ± standard deviation.

## 3. Results

### 3.1. Characteristics of Participants

There was no statistically significant differences in age, height, weight, BMI, or L4/L5 disk height between the two groups (Table 1).

### 3.2. MRI Findings 

In the spondylolisthesis group, 19 patients (86.3%) had L4/L5 or L5/S1 spondylolisthesis, and 29 patients (96.6%) had grade I spondylolisthesis (Table 2). Disk degeneration and L4/L5 facet arthropathy grades were measured, there were no statistically significant differences in disk degeneration and facet arthropathy grade between two groups (Table 3).

For the MF muscle, the mean total CSA, rCSA, rFCSA, and SMI of total CSA showed no statistically significant difference between the two groups. However, the mean FCSA of the MF was significantly smaller in patients with DLS (244.63 vs. 298.15 mm^2^; *p* = 0.030). The mean FCSA: total CSA ratio (43.67 vs. 55.34%; *p* = 0.001) showed a statistically significant difference between two groups (Cohen’s Effect size d: 0.876, power: 0.924). This means, the result showed more fat infiltration of MF in patients with DLS (56.33 vs. 44.66%; *p* = 0.001). The mean SMI of the FCSA (107.44 vs. 129.79 mm^2^/m^2^; *p* = 0.38) were significantly lower in patients with DLS than in patients without DLS. 

For the ES muscle, there were no statistically significant differences between two groups in the mean total CSA, rCSA, rFCSA, ratio of FCSA: total CSA, and SMI of total CSA. However, the mean FCSA (783.33 vs. 666.22 mm^2^; *p* = 0.028) and SMI FCSA (343.95 vs. 288.78 mm^2^/m^2^; *p* = 0.012) of ES muscle was a statistically larger in the patients with DLS.

There were no statistically significant differences in MF total, CSA: ES total, and CSA ratio between the two groups. However, the mean MF FCSA: ES FCSA ratio showed a statistically significant difference between the two groups (0.33 vs. 0.45; *p* < 0.001) (Table 4). Patients with DLS showed much smaller MF FCSA to ES FCSA ratio. 

After applying logistic regression analysis, all variables, except rFCSA (*p* = 0.822), correlated significantly with patients with DLS. Among five sets of variables, MF FCSA/total CSA with ES FCSA/total CSA showed largest AUC area (0.827).

The ES FCSA / total CSA was an independent predictor of lumbar spondylolisthesis (OR = 1.092, *p* = 0.016), while the MF FCSA/total CSA was an independent protective factor (OR = 0.898, *p* = 0.002) (Table 5).

## 4. Discussion

In this study, patients with radiculopathy showed no significant difference in total CSA of the MF or ES. Instead, the fat infiltration ratio (100-FCSA /total CSA%) and functional CSA of the MF were significantly different between those with and without DLS. The patients with DLS showed more fat infiltration in multifidus muscle and small FCSA. In contrast, erector spinae muscle showed a large FCSA in patients with DLS. This result is consistent with that of a previous study on fat infiltration. Fat infiltration is a sign of muscle atrophy, and the replacement of muscle with fat, while changing the function of the muscle, may not significantly alter its CSA [21].

Several studies have investigated the association between spondylolisthesis and paraspinal muscle CSA [22]. In a computer tomography (CT) study, a significant association was found between the density of the multifidus muscle at level L4 and spondylolisthesis [23]. A previous Chinese study evaluated the MRIs of 149 middle-aged patients with degenerative spondylolisthesis and age- and sex-matched controls. The MF muscle atrophy ratio (lean CSA: total CSA) of the study group tended to be significantly lower than that of the control group, whereas the signal intensity of the paraspinal muscles and the ES muscle atrophy ratio were significantly higher in the former than in the latter [19]. 

Thakar et al. assessed the CSA of lumbar paraspinal muscles in 120 adults with isthmic spondylolisthesis. The mean CSA value of the ES was significantly higher in the study cohort (*p* = 0.002) than in age- and sex-matched controls, whereas the CSA of the MF muscle was significantly lower (*p* = 0.009) [19].

In this study, patients with DLS showed high mean ES FCSA and SMI FCSA. This means that larger lean muscle volume of erector spinae was found in patients with DLS. We can cautiously conclude that patients with spondylolisthesis suffer from segmental atrophy of the MF muscle. ES hypertrophy may compensate for this instability [22].

The MF muscle is divided into five myotomes, each innervated by a single spinal segment. Muscle fibers attached to the spinous process of a particular vertebra are segmentally innervated by the medial branch of the dorsal ramus, which originates inferior to the respective vertebra [19]. The deep fiber of the MF is composed of type I (slow-twitch) fibers. It is suitable for low-load tonic activity. It is more vulnerable to immobilization or pain than type II (fast-twitch) fibers and can be considered as one of the reasons for selective atrophy of MF [19].

This study compared various measures. The fat infiltration ratio and rFCSA was investigated in patients with DLS in previous studies; SMI, MF:ES ratio were first applied in this study. However, to our knowledge, this is the first study to compare these variables simultaneously. Furthermore, we attempted to find the best variable to predict patient with DLS. In this study, increased fat infiltration in the multifidus muscle with decreased fat infiltration in the erector spinae muscle are the best predictors of the patient with DLS. 

A strength of our study was the diagnosis of lumbosacral radiculopathy. Most of the previous studies used age- and sex-related normative populations as control groups, failing to exclude the influence of chronic radiculopathy-induced MF atrophy. Only our study included electrodiagnostic testing and patients with radiculopathy who were older than 65 years participated in this study. Furthermore, we used quantitative methods to accurately measure fat infiltration, as opposed to a previous study. 

Our study provides a basis for spinal stabilization exercise and physiotherapy. The MF muscle is an important stabilizer of the lumbar spine neutral zone, and atrophy of the muscle decreases the ability to control the neutral zone and is strongly associated with LBP [24]. Spinal stabilization has been observed to be more effective over time in treating LBP than minimal intervention and exercise therapy alone. Therefore, the treatment focus has shifted to reactivation and strengthening of the smaller muscles of the spine to improve long-term stabilization of the vertebral column [24]. The benefit of targeted MF muscle strengthening programs utilizing this functional peculiarity has been demonstrated in patients with acute and chronic LBP with DLS. A study of subjects with acute back pain demonstrated normalization of the MF areas within a month after a specific motor re-education exercise program [11].

This study had several limitations to this study. First, the sample size was small, and only women were included in this study. Second, this was cross-sectional study and performed retrospectively. For future prospective studies, longitudinal data are needed to determine whether DLS is a cause or result of replacement of MF muscles with fat. Third, a standardized assessment of pain severity and daily activity was not performed. Medical treatment that leads to pain relief may also affect the structure of the lumbar muscles; this should be examined in future prospective studies. 

## 5. Conclusions

Small FCSA of the multifidus muscle with high degree of fat infiltration and large FCSA of the erector spinae muscle were found in the DLS group with chronic radiculopathy.

MF FCSA/ES FCSA, SMI FCSA, ratio of FCSA to total CSA, and FCSA were independent predictors of lumbar spondylolisthesis, while increased fat infiltration in the multifidus muscle with decreased fat infiltration in erector spinae muscle best predict the patient with DLS.

## Figures and Tables

**Figure 1 ijerph-18-04037-f001:**
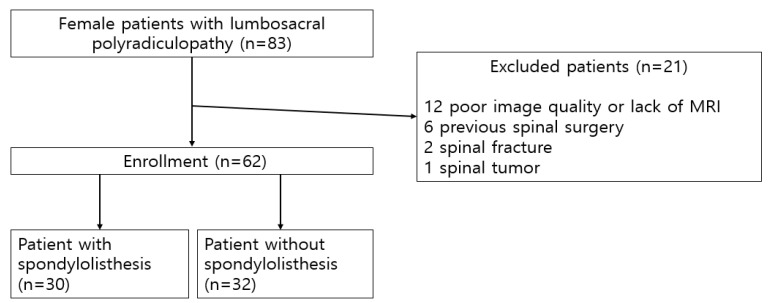
Flow chart of patient enrollment and grouping.

**Figure 2 ijerph-18-04037-f002:**
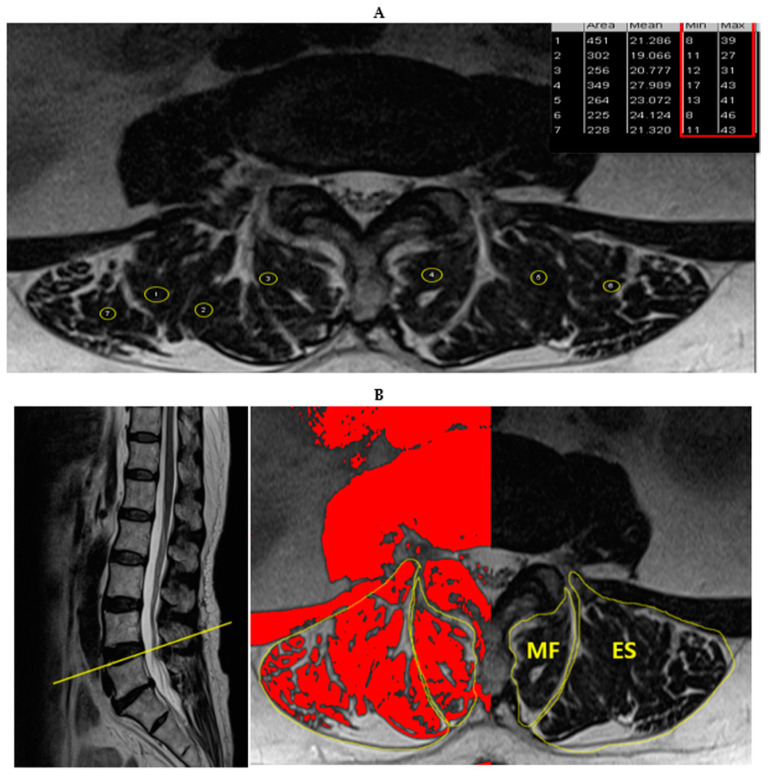
Quantitative paraspinal muscle measurement. Measurement of cross-sectional area (CSA) and functional CSA of the multifidus muscle (MF) and erector spinae (ES) muscle at the L4-L5 disk level. (**A**) Sample selection of regions of interest (ROIs) to define upper and lower signal intensity threshold limits. (**B**) Sagittal and axial T2 weighted MR image of the FCSA of the paraspinal muscle group (red area), using the threshold method with ROI tracing.

**Table 1 ijerph-18-04037-t001:** Demographic and radiological characteristics of patients.

Sex	Patients with Spondylolisthesis (n = 30)	Patients without Spondylolisthesis (n = 32)	*p*
Age (years)	74.1 ± 5.6	74.5 ± 5.4	0.727
Height (cm)	150.6 ± 6.4	151.7 ± 6.3	0.511
Weight (kg)	56.8 ± 7.1	57.3 ± 11.0	0.861
BMI (kg/m^2^)	25.0 ± 2.6	24.8 ± 4.1	0.801
L4-L5 disc height (mm)	9.4 ± 2.7	9.47 ± 2.8	0.949

**Table 2 ijerph-18-04037-t002:** Radiologic features spondylolisthesis group.

Radiologic Features	Number of Patients (%)
Level of listhesis	
L3-4	2 (6.7%)
L4-5	19 (63.34%)
L5-S1	9 (30.0%)
Listhesis grade	
grade 1	29 (96.6%)
grade 2	1 (3.3%)
grade 3	0
grade 4	0

**Table 3 ijerph-18-04037-t003:** Radiologic features of 62 patients.

	Patients with Spondylolisthesis (n = 30)	Patients without Spondylolisthesis (n = 32)	*p*
Disc degeneration grade			
1	0	0	
2	0	0	
3	2	1	0.92
4	14	16	
5	14	15	
L4/L5 facet arthropathy grade			
0	0	0	
1	0	3	
2	11	16	0.07
3	19	13	

**Table 4 ijerph-18-04037-t004:** Magnetic resonance imaging findings for multifidus and erector spinae muscle groups by degenerative lumbar spondylolisthesis.

Muscle Group	Patients with Spondylolisthesis (n = 30)	Patients without Spondylolisthesis (n = 32)	*p*
Multifidus muscle group			
Total CSA (mm^2^)	553.80 ± 136.59	556.25 ± 157.27	0.948
FCSA (mm^2^)	244.63 ± 101.39	298.15 ± 88.48	0.030
Ratio of FCSA to total CSA (%)	43.67 ± 13.33	55.34 ± 13.31	0.001
CSA/VB (rCSA)	0.473 ± 0.395	0.365 ± 0.156	0.162
FCSA/VB (rFCSA)	0.202 ± 0.203	0.199 ± 0.097	0.940
SMI(mm^2^/m^2^)			
of Total CSA	244.09 ± 60.56	242.26 ± 68.94	0.912
of FCSA	107.44 ± 43.91	129.79 ± 39.11	0.038
Erector spinae muscle group			
Total CSA (mm^2^)	1511.05 ± 273.26	1384.7 ± 281.06	0.078
FCSA (mm^2^)	783.33 ± 225.09	666.22 ± 182.28	0.028
Ratio of FCSA to total CSA (%)	51.69 ± 11.79	48.57 ± 10.94	0.283
CSA/VB (rCSA)	1.225 ± 0.975	0.879 ± 0.261	0.058
FCSA/VB (rFCSA)	0.643 ± 0.586	0.432 ± 0.180	0.057
SMI(mm^2^/m^2^)			
of Total CSA	667.11 ± 124.70	604.71 ± 128.65	0.057
of FCSA	343.95 ± 94.06	288.78 ± 72.23	0.012
MF CSA/ES CSA	0.37 ± 0.98	0.45 ± 0.23	0.062
MF FCSA/ES FCSA	0.33 ± 0.15	0.45 ± 0.12	0.000

CSA, cross-sectional area; FCSA, functional CSA; SMI, skeletal muscle index; VB, vertebral body; MF, multifidus; ES, erector spinae.

**Table 5 ijerph-18-04037-t005:** Logistic regression analysis: factors associated with the occurrence of degenerative lumbar spondylolisthesis

	Significance	Odd Ratio	95% CI for Odd Ratio		AUC Area
			Lower	Upper	
MF FCSA/ES FCSA	0.006	0.001	0	0.115	0.796
SMI MF FCSA	0.015	0.974	0.954	0.995	0.827
SMI ES FCSA	0.002	1.017	1.006	1.027	
MF FCSA/total CSA (%)	0.002	0.898	0.839	0.961	0.841
ES FCSA/total CSA (%)	0.016	1.092	1.107	1.174	
MF FCSA (mm^2^)	0.005	0.985	0.975	0.996	0.817
ES FCSA (mm^2^)	0.004	1.007	1.002	1.012	
MF FCSA/VB (rFCSA)	0.882	1.297	0.042	39.89	0.758
ES FCSA/VB (rFCSA)	0.022	71.798	1.835	2809.979	

CSA, cross-sectional area; FCSA, functional CSA; SMI, skeletal muscle index; MF, multifidus; ES, erector spinae.

## Data Availability

The data presented in this study are available on request from the corresponding author. The data are not publicly available due to privacy matters.

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
