# Peer review of "Association of Lumbar Paraspinal Muscle Morphometry with Degenerative Spondylolisthesis"

_ijerph, 2021, doi:10.3390/ijerph18084037_

Round 1
Reviewer 1 Report
This paper claims that the functional CSA (signal of muscle tissue) is lower in patients w/ spondylolisthesis than w/o spondylolisthesis, while total CSA in lumbar radiculopathy patients w/ or w/o spondylolisthesis are consistent. The result suggests the fat tissue replaces the muscle tissue in multifidus more severely in those radiculopathy patients w/ spondylolisthesis than those w/o spondylolisthesis. Therefore, this interesting observation supports the hypothesis that the strong muscle is key to maintain the vertebral structure and avoid spondylolisthesis. Overall, the observation is interesting, but it is obvious that radiculopathy and spondylolisthesis may also cause muscle tissue loss, because the patients may lose some ability and decrease their energy spend. Thus, it is still an open question of whether muscle loss causes spondylolisthesis.
Major concerns:
1, CSA loss was found in many radiculopathy patients. Is the data in this research consistent with the published data before, that means the 55 patients in this study have comparable CSA loss with data before? The percentage of CSA loss and FCAS loss in patients in this research compared with the normal person may help us estimate the severity of FCAS loss in patients with both radiculopathy and spondylolisthesis.
2, the quantification of FCSA and total CSA is not well described in the paper. Spondylolisthesis may also affect the total CSA and FCSA dramatically in the spondylolisthesis position. The method part did not discuss how the quantification from spondylolisthesis position has been dealt with. It is unclear whether CSA and FCSA in the spondylolisthesis position are consistent with the other parts. It is possible that CSA and FCSA in spondylolisthesis position might be different from the CSA and FCSA of the other part and directly mislead the major claim of the paper.
3, The logistic regression is not well documented. A curve of logistic regression with the proper explanation will make the paper more understandable.
Minor concerns:
Is BMI the abbreviation of body mass index? Please clarify it first then use it.
Is “poas muscle” psoas muscle?
Please add citations to support the claim of lumber muscle change in patients of radiculopathy, line 53-54.
Line 56-57, the sentence “Thus, it can be said…………, and chronic radiculopathy” needs to be revised. The sentence also needs some published literature to support it.
The fat infiltration mentioned in line 58 also needs citation to support.
The sentence “However, when DLS patients have chronic radiculopathy……MF atrophy” is unclear. Please revise it.
Reviewer 2 Report
The study describes the findings coming from limited amount of experimental data, since the research is limited to one technique (MRI). The group of patients is too small to use the results to construct i.e. valid diagnostic tool. The results are interesting, because show the direction for further treatment. As a reader I appreciate that authors underline limitations of their study, however the limitations are gross. Therefore in my opinion more diverse techniques should be used (also on bigger group, including males) as well as better control of the experiment should be performed to confirm results.Author Response
Please see the attachment.

Reviewer 3 Report
The authors presented a retrospective study comparing various measures of cross-sectional area of the paraspinal and multifidus muscles via MRI among patients with and without DSL. The study is unique and would contribute to the literature by further describing characteristics of DSL and deriving future directions for rehabilitation. Although the study has merit, in the current form the research report must be revised prior to being considered suitable for publication. Below the authors will find some specific comments, but my major concerns are:
1) There is limited justification for the included dependent variables. What does each variable actually represent and how does it differ from another.
2) The authors discuss fatty infiltration, but do not describe which variable(s) correspond to fatty infiltration. I am assuming that the conclusion of an increased fatty infiltration stems from lack of difference in total CSA; however, differences in FCSA between the groups.
3) The justification for the sample size is not included
4) The interpretation of the linear regression and associated ORs are lacking.
Without further clarification of the above points, it is difficult to fully interpret the results of this study.
Specific Comments
Abstract:
Line 21: DLS is not defined in the abstract
Lines 26-27: Include the specific OR
Introduction:
The introduction does not adequately justify the need for this study. Currently, the organization of the introduction limits the connection between the background of the pathology, the characteristics of the pathology, the common pathophysiology/mechanics of the pathology, and proposed study. I urge the authors to re-write the introduction to better funnel the information to lead to a well-defined purpose statement. Below is a general guide to organizing the introduction:
1st Paragraph: General background of DLS. This paragraph should include what it is, how common it is, and the consequences.
2nd Paragraph: Pathomechanics of the DLS and other characteristics.
3rd Paragraph: The aging process and consequences to muscle and how that may contribute to DLS
4th Paragraph: Describe the importance of the lumbar paravertebral muscles and the importance of this muscle group as it related to DLS. Why study these muscles when other muscles (MF) have already been identified as altered and contributors of DLS?
5th Paragraph: Purpose statement and hypotheses. Define and justify your patient population. Why did all need to have polyradiculopathy?
Line 73: What is meant by ‘presence of L-spine’
Line 89: Please clarify the before or after 6-weeks of diagnosis? Was this accounted for in the statistical analysis as 6 weeks is enough time for morphological changes in muscle to occur.
Line 108: Was the assessor blinded to group assignment?
Line 118: The authors should operationally define each variable: CSA, FCSA, rCSA, rFCSA, ratio of FCSA. What unique information do each of these variables provide?
Line 128: The authors should consider calculating Cohen’s d effect sizes and associated 95% confidence intervals to determine the magnitude of differences between groups for each variable.
Line 129: How did the authors decide the independent variables and why were these variables not included in the t-test analyses.
Line 129: Were the authors adequately powered to include this number of variables?
Line 162: What are the specific odd’s ratios?
Reviewer 4 Report
This paper examined muscle morphometry in patients with lumbar radiculopathy with and without DLS. Overall, this manuscript is well written but there are too many abbreviations to keep track of even for readers familiar with the majority of them. Please consider removing abbreviations that are infrequently used from the manuscript.
Abstract
In the final sentence of the abstract, why not include the actual OR? Just stating > or < 1 does not provide the reader much information on interpreting the predictive or protective effect of the measure.
Introduction
Lines 36-39: This sentence is awkwardly written and should be revised for clarity and proper sentence structure.
Line 39: Consider moving this sentence to the end of the paragraph for better flow from pathogenesis to causative factors to symptoms.
Lines 44-45: Is this increased attention on the paraspinals specific to patients with DLS? Please clarify for the reader.
Lines 48-50: Please provide citations for these statements.
Line 56: No evidence was provided in the previous paragraph regarding women. Please add that evidence or remove this statement.
Line 58: Provide a citation for this statement.
Line 59: Do the authors mean deficit instead of defect?
Line 60: Change your wording throughout the manuscript to be patient centered. For example, refer to patients with DLS not DLS patients.
Line 61: I suggest revision of this sentence referring to the cause of MF atrophy considering this is a retrospective study and causation cannot be determined.
Methods
Was any sort of sample size estimate conducted?
Please select 1 spelling of disc and use it consistently throughout the manuscript.
There is a lot of detail to the methods, but not much regarding how the actual CSA measures were performed. Simply citing previous research is not sufficient.
Lines 73-74: The 3rd inclusion criterion is unclear “the presence of L-spine on MRI”. The way it reads to me the authors were concerned only that participants had a lumbar spine. Please revise for clarity.
Line 91: Who performed MRI gradings? What is the reliability for this person or these individuals? Additionally, later in this section please address the reliability of the investigator determining the various CSA measurements.
Line 127: Was rFCSA included in the statistical analysis?
Results
Lines 135-136: Please remove data from the text if included in the table.
Lines 141-145: Were these findings based on observation (counts of persons in each category) or were these statistically significant differences?
Line 153: This sentence can be removed as it was stated at the beginning of the paragraph.
Line 157: I suggest including this finding with the previous sentence listing the results that were not statistically significant rather than giving it its own sentence.
Discussion
It is rather unconventional to lead off the discussion section talking about previous work and for 2 paragraphs not relate it back to the present study. I suggest revising the discussion section to focus on your findings first and then relate them to previous literature.
Line 188: I appreciate the cautious conclusion, but I wonder if it is still overstating the findings since all levels of the MF were not measured for atrophy.
Lines 197-203: These sentences state the same information. Please remove the redundancy.
Line 212: This study does not support the benefits of any therapy as no therapy was evaluated.
Lines 225-226: This sentence needs to be reworded to make it clear that the retrospective, cross-sectional nature of this study is the limitation, and the longitudinal data would be the future direction.
Conclusion
This section is merely a restatement of the findings and no take home message is provided. Please include the key take home message for the reader.
Tables and Figures
Figure 1: Under the reasons for exclusion from the study, the numbers of persons excluded adds up to 22 not 21. Please clarify the reason for the discrepancy.
Table 1: Weight should be mass. Units for BMI and disc height should be provided.
Round 2
Reviewer 2 Report
Thank you for in depth corrections and additional data.
Reviewer 3 Report
I thank the authors for their efforts to address each of my previous comments. The manuscript has greatly improved. My only remaining comment is to complete a thorough review of the text as numerous grammatical errors remain. Other than that, great work on this project.